# A Parameter-Free Outlier Detection Algorithm Based on Dataset Optimization Method

**Liying Wang, Lei Shi \*, Liancheng Xu, Peiyu Liu, Lindong Zhang and Yanru Dong**

School of Information Science and Engineering, Shandong Normal University, Jinan 250358, China;
2017020848@stu.sdnu.edu.cn (L.W.); lchxu@163.com (L.X.); liupy@sdnu.edu.cn (P.L.);
2018020898@stu.sdnu.edu.cn (L.Z.); answer3210@163.com (Y.D.)
\* Correspondence: sdnushilei@163.com; Tel.: +86-159-5314-2953

**Abstract:** Recently, outlier detection has widespread applications in different areas. The task is to identify outliers in the dataset and extract potential information. The existing outlier detection algorithms mainly do not solve the problems of parameter selection and high computational cost, which leaves enough room for further improvements. To solve the above problems, our paper proposes a parameter-free outlier detection algorithm based on dataset optimization method. Firstly, we propose a dataset optimization method (DOM), which initializes the original dataset in which density is greater than a specific threshold. In this method, we propose the concepts of partition function (P) and threshold function (T). Secondly, we establish a parameter-free outlier detection method. Similarly, we propose the concept of the number of residual neighbors, as the number of residual neighbors and the size of data clusters are used as the basis of outlier detection to obtain a more accurate outlier set. Finally, extensive experiments are carried out on a variety of datasets and experimental results show that our method performs well in terms of the efficiency of outlier detection and time complexity.

**Keywords:** dataset optimization method; partition function; threshold function; parameter-free outlier detection

## 1. Introduction

Outlier detection is an important branch of data mining, aiming at finding noise data or potential information. In recent years, with the development of information technology and machine learning, outlier detection has received more and more attention in different applications such as intrusion detection, image processing, medical research, etc. [1–6].

Outliers are a form of data existence that deviates from other observations. In many areas, outliers play a more important role than ordinary data points because they may show different forms from normal patterns and cause damage to users [7]. At present, many researchers have proposed many outlier detection algorithms [8], which include the distribution-based method, depth-based method, distance-based method, density-based method and so on [9–12]. However, many outlier detection algorithms have their advantages and disadvantages and they cannot be fully applicable to all detection algorithms [13]. For example, a distribution-based outlier detection algorithm is devised by judging whether the data object deviates from the standard distribution to determine whether the object is an outlier. However, the distribution-based method usually has unknown data distribution. The depth-based method can solve the problem that the distribution of data objects is unknown and difficult to detect, but for more than three-dimensional data space, the detection effect is not ideal. A distance-based outlier detection algorithm can solve this problem, but the distance-based method cannot solve the effect of local detection. The density-based method can solve the problem of local

detection. However, both distance-based and density-based outlier detection methods are based on the selection of nearest neighbors and parameters. So it is very sensitive to the choice of the nearest neighbors and parameters. Additionally, many outlier detection algorithms need parameters to specify the number of outliers and the percentage of outliers in the dataset, but this is not easy [14,15].

Simultaneously, in the process of outlier detection, each data point exists in different forms, which includes large data clusters, small data clusters, and sparsely distributed outliers. The proportion of outliers is so small, the percent of it is around 5% and even less [6]. The traditional outlier detection algorithm needs to test each data point in the dataset based on distance, density, and distribution, resulting in the low efficiency of the outlier detection algorithms. Therefore, it is very important to initialize the dataset.

To further solve the above problems, our paper proposes a parameter-free outlier detection algorithm, solving the above problems efficiently. The main contributions are as follows.

(1)　Dataset optimization method (DOM) is proposed. In our method, we propose the concepts of partition function (P) and threshold function (T). It is used to initialize the original dataset and filter the data in which density is greater than a specific threshold. Our method greatly reduces the data size of the algorithm and improves the efficiency of the algorithm.

(2)　A parameter-free detection method is established. In this method, we propose the concept of the number of residual neighbors to further optimize the algorithm based on mutual neighbors. The number of residual neighbors and the size of data clusters are used as the basis for calculating the anomalous degree of data points, and the parameter-free detection is carried out. The advantage is that it reduces the sensitivity of the nearest neighbor parameter selection and achieves a more accurate outlier set.

The structure of this paper is as follows. The Section 2 is about related work. The Section 3 introduces the outline and detailed information about the parameter-free outlier detection algorithm model in our paper. The Section 4 introduces the experimental results and compares our method with other methods. In Section 5, we propose a conclusion about our method.

## 2. Related Works

### 2.1. Definition of Outliers

Outlier detection is an important aspect of data mining. The purpose of outlier detection is to detect data points that are inconsistent with most data points in the dataset. The definition of outlier [16] is different according to different detection methods, but Hawkins [17] gives the classic definition of an outlier: an outlier is a data point which is very different from other data points. Therefore, we have the suspicion that the normal data points and the outliers are generated by different mechanisms. The suspicion is widely accepted by more and more researchers and has become the basis of the study about outliers.

### 2.2. The Density-Based Outlier Detection Algorithm

At present, many researchers have proposed many outlier detection algorithms [8], which include the distribution-based method, depth-based method, distance-based method, density-based method and so on. By analyzing the characteristics of the above traditional outlier detection algorithms, we find that the density-based outlier detection algorithm is the classic traditional outlier detection algorithm [18]. Therefore, the algorithm proposed in this paper is mainly compared with the density-based outlier detection algorithm.

Density-based outlier detection algorithm is a very classical anomalous data mining algorithm [19,20], mainly by comparing the density of each point P and its neighboring points to determine whether the point is an outlier; and if the density of point P is lower, it is more likely to be identified as an outlier. The

density is calculated by the distance between points. The farther the distance between points is, the lower the density is. The closer the distance is, the higher the density is. The following definitions are involved.

D is the dataset. $p$ and $q$ are some objects in the dataset. K is the nearest neighbor. $d(p,q)$ denotes the distance between two points $p$ and $q$.

**Definition 1.** *K-distance: the distance from the k-th point of data point P, excluding p.*

**Definition 2.** *K-distance neighborhood of P: the k-th distance neighborhood of point P is all points within the k-th distance of P, including the k-th distance.*

**Definition 3.** *Reach distance: the k-th reachable distance from point o to point p is defined as*

$$reach - distance_k(p,o) = max\{k - distance(o), d(p,o)\}. \tag{1}$$

*That is, the k-th reachable distance from O to P, at least the k-th distance of O, or the real distance between O and P.*

**Definition 4.** *Local reachability density: the local reachability density of point P is expressed as*

$$lrd_k(p) = 1/(\frac{\Sigma_{o \in N_k(p)} reach - dist_k(p,o)}{|N_k(p)|}. \tag{2}$$

**Definition 5.** *Local outlier factor: the local outlier factor of point P is expressed as*

$$lof_k(p) = \frac{\Sigma_{o \in N_k(p)} \frac{lrd(o)}{lrd(p)}}{|N_k(p)|} = \frac{\frac{\Sigma_{o \in N_k(p)} lrd_k(o)}{|N_k(p)|}}{lrd_k(p)}. \tag{3}$$

The density-based outlier detection algorithm determines the outlier subset by calculating the outlier factor value for each data point. In general, several data points with large outlier factors may be outliers. When the dataset is small, this outlier detection method has high detection efficiency. However, the density-based outlier detection algorithm also has disadvantages:

1. When we apply the outlier detection algorithm to detect the dataset, most data points are normal points, and the number of outliers is small. When the dataset is large, the algorithm needs to calculate the outlier factor value for each data point, which means that there are a lot of calculations in the outlier detection algorithm, so the time complexity of the algorithm is high. Therefore, filtering the dataset to reduce the interference of normal data is particularly important.
2. The traditional density-based outlier detection algorithm needs to repeatedly calculate the k-distance to obtain the local outlier factor value. The accuracy of the algorithm depends on the selection of the nearest neighbor parameter, so it is very sensitive to the parameter selection.

*2.3. Parameter Selection in Outlier Detection*

Proper parameter selection is very important for the outlier detection algorithm. Many improved outlier detection algorithms intuitively give the size of the nearest neighbor parameter k, but where the parameters come from is ambiguous. To improve the accuracy of the algorithm and reduce the influence of parameter selection, many outlier detection algorithms have been proposed [21–25]. Ha et al. proposed a new outlier detection method including iterative random sampling. The observable factor is used to solve the influence of parameter k, but the algorithm is affected by the number of iterations. Rahman et al. realized parameter-independent clustering and outlier detection by using two

new concepts of unique nearest neighbor and unique neighborhood set. However, the time efficiency of the algorithm is low. Zhu et al. proposed a self-adaptive neighborhood method without parameter k based on a weighted natural neighborhood graph. In the above method, firstly, a natural structure is created to find global outliers and outlier clusters, and then natural outlier factors are calculated to find local outliers. Additionally, a new instance reduction algorithm is proposed. This method uses the concept of natural neighborhood. The algorithm eliminates global outliers without user-defined parameters and reduces the impact of outliers on the boundary. However, these algorithms do not take both efficiency and parametric testing into account.

*2.4. Nearest Neighbor*

The idea of the nearest neighbor plays an important role in data mining and machine learning. The main roles are in classification, clustering, and outlier detection [26–28]. At present, the most commonly used nearest neighbor thoughts are k-nearest neighbor thoughts and r-nearest neighbor thoughts. The idea of k-nearest neighbor is centered on a data point p, and k-nearest neighbors forms the nearest neighborhood of the data point p, which is called k-nearest neighbor of point p. The idea of r-nearest neighbor is to take a data point p as the center of the circle and use as the radius. All data points within the radius of rare regarded as the nearest neighbor of the data point p, which is called the r-nearest neighbor of point p. The nearest neighbor can reflect the relationship between data points, and one form of this relationship is the nearest neighbor graph [29]. Neighborhood graphs can be constructed by connecting each data point to its nearest neighbor. As the number of nearest neighbors k increases, the nearest neighbors and the nearest neighbor graph of data points change correspondingly. With the wide attention and research of outlier detection, Brito et al. proposed a clustering and anomalous detection algorithm based on the mutual k-nearest neighbor graph [30]. Huang et al. proposed a new outlier clustering detection algorithm [31]. Based on the concept of the mutual neighbor graph, it can accurately detect the abnormal values and clusters of abnormal values without parameter top-n.

Compared with the traditional outlier detection algorithm, the parameter-free outlier detection algorithm proposed in our paper mainly solves two main problems:

1. The traditional outlier detection algorithm needs to calculate the outlier value of each data point, which leads to high time complexity of the algorithm. Based on the efficiency of outlier detection, our paper proposes a dataset optimization method to improve the efficiency of the algorithm.
2. The traditional outlier detection algorithm needs to repeatedly calculate the k-distance to get local outliers, which is very sensitive to parameter selection. There are many improved outlier detection algorithms that intuitively give the size of the parameters and without specific analysis of the source of the parameters. Based on the nearest neighbor, our paper proposes the concept of the number of remaining neighbors. The accuracy of parameter selection is improved.

## 3. Methods

In order to improve the efficiency of outlier detection and avoid the influence of the nearest neighbor parameters on the accuracy of the algorithm, we propose a parameter-free outlier detection algorithm. The parameter-free outlier detection algorithm mainly consists of two stages. Firstly, we propose a DOM method to initialize the original data. In this method, we propose the concepts of partition function (P) and threshold function (T); and filtering the data clusters in which density is greater than a specific threshold, we obtain the initialized datasets. Then, we establish a parameter-free detection method, in which we introduce the concept of the number of residual neighbors and determine outliers via observing the changes of the number of residual neighbors of data points and the size of data clusters.

*3.1. Dataset Optimization Method*

Each data point exists in different forms, including large data clusters, small data clusters, and sparsely distributed outliers. The traditional outlier detection algorithm needs to calculate the outlier value of each data point, which leads to high time complexity of the algorithm.

As shown in Figure 1, the blue data points are C1 dense clusters, and the blue-green data points are C2 sparse clusters. They are normal data clusters in the dataset. Red data points are outliers. The density-based outlier detection algorithm needs to calculate the outlier degree of each data point (C1 cluster, C2 cluster, outlier), resulting in a high time complexity for the algorithm.

According to the distribution characteristics of outliers and the problem of high time complexity in detection, our paper proposes a dataset optimization method (DOM). In the dataset optimization method, we propose two new functions: partition function (P) and threshold function (T), which are used to determine the data space partition and threshold, and finally achieve the effect of initializing the dataset without parameters. As shown in Figure 1, the large data cluster (C1) is filtered by the DOM, which ensures that the amount of data entering the next stage is small. On the other hand, it improves the efficiency of the algorithm and requires low time complexity.

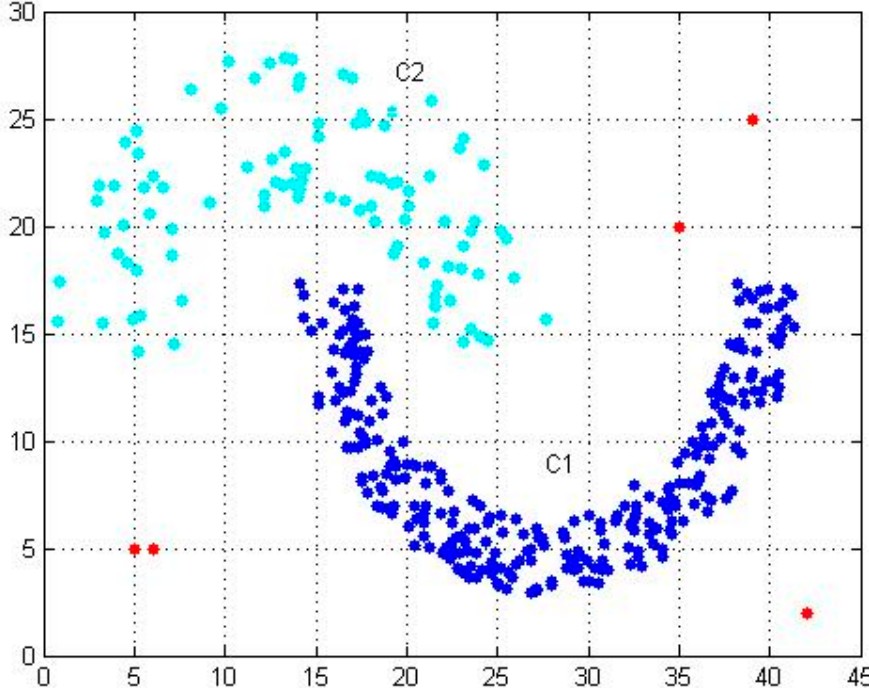

**Figure 1.** Existence forms of data (blue normal points are C1 and that in blue-green are C2, the outliers points are red).

To describe the process of dataset optimization, we use Algorithm 1 to describe two-dimensional datasets. In Algorithm 1, we first input the original dataset. Then, in step 1, we use the value of partition function to complete spatial partition and regional data point statistics. In step 2, we compare the number of data points and the size of the threshold function for each region and then determine whether each region is dense or non-dense. If the number of data points in the region is larger than the threshold function, the region is dense. We initialize the data points in this area. Otherwise, the area is not dense. We preserve non-dense data points in the dataset. Finally, we completed the dataset optimization method to obtain candidate datasets as follows:

---

**Algorithm 1.** Dataset optimization method

---

input: Original Dataset {O}
output: Candidate Dataset{C}
initialize: {c} = ∅
  step1:
    use function Split Area() split space area
    Block Num = Split Area({o})
      //block Num is the number of Blocks
    for I = 1, 2, . . . block Num do
      Num = Count Point Num()
    End for
  step2:
    initialize threshold T
    for I = 1, 2, . . . block Num, do
      get current block points: { block I }
      get the number num of current block
      if(num<T)
        this block area is non-dense set
        {c} = {c}∪{ block I }
      else
        { block I } = ∅
      endif
    endfor

---

We propose two new functions, partition function, and threshold function, to partition the data space and achieve the effect of initializing datasets without parameters.

The partition function determines the partition of space and divides N data points into different regions according to their respective positions. If the dataset is large, the partition area should be relatively large. Otherwise, since the number of partition areas are so few that the data in each partition is too dense, the outliers belonging to the abnormal areas are wrongly distributed to the normal area. When the dataset is small, the partition area should be relatively small.

Threshold function is a measure to evaluate whether the areas in the partition function are dense areas. When the number of data points in each region is greater than the threshold function, the data cluster is dense. Then, we filter such dense data clusters to obtain the initialized candidate datasets.

In order to determine the reasonable partition function and threshold function, we verify the data size of 1–100,000 datasets. We express it mathematically as follows.

$N = \{o_1, o_2, \ldots, o_j, \ldots o_n\}$ denotes the dataset; $|N|$ is the number of N; P is the number of rows or columns partitioned by regions; and T refers to the size of threshold function.

The partition function is used to partition the dataset. If the dataset is large, the partition area should increase accordingly. If the dataset is small, the partition area should be correspondingly smaller. Figure 2 shows the distribution of datasets with $|N|$ = 2000 when the partition function is 18. Figure 3 shows the distribution of datasets with $|N|$ = 2000 when the partition function is 7. When the partition function is 7, the dataset of each region is too dense, and the possible abnormal data points are wrongly divided into a region with normal points. Therefore, the size of the partition function and threshold function should change with the change of dataset. Therefore, we propose the definition of partition function P.

$$P = |N|^{1/3} + |N|^{1/4} \tag{4}$$

| 0 | 0 | 0 | 0 | 0 | 0 | 0 | 0 | 0 | 0 | 0 | 0 | 0 | 0 | 0 | 0 | 0 | 0 |
|---|---|---|---|---|---|---|---|---|---|---|---|---|---|---|---|---|---|
| 0 | 0 | 0 | 7 | 10 | 19 | 15 | 8 | 0 | 0 | 0 | 0 | 0 | 0 | 0 | 0 | 0 | 0 |
| 0 | 0 | 6 | 19 | 18 | 19 | 23 | 16 | 26 | 3 | 0 | 0 | 0 | 0 | 0 | 0 | 0 | 0 |
| 0 | 0 | 21 | 32 | 30 | 25 | 24 | 24 | 21 | 13 | 0 | 0 | 0 | 0 | 0 | 0 | 0 | 0 |
| 0 | 9 | 21 | 21 | 22 | 21 | 30 | 17 | 17 | 19 | 3 | 0 | 0 | 0 | 0 | 0 | 0 | 0 |
| 0 | 14 | 28 | 33 | 24 | 19 | 31 | 21 | 20 | 21 | 6 | 0 | 0 | 0 | 0 | 0 | 0 | 0 |
| 1 | 20 | 17 | 24 | 12 | 30 | 19 | 18 | 16 | 17 | 9 | 0 | 0 | 0 | 0 | 0 | 0 | 0 |
| 0 | 19 | 22 | 15 | 13 | 21 | 29 | 24 | 20 | 29 | 19 | 0 | 0 | 0 | 0 | 0 | 0 | 0 |
| 0 | 14 | 25 | 24 | 20 | 21 | 26 | 26 | 21 | 14 | 8 | 0 | 0 | 0 | 0 | 0 | 0 | 0 |
| 0 | 11 | 19 | 30 | 28 | 19 | 18 | 24 | 16 | 29 | 6 | 0 | 0 | 0 | 0 | 0 | 0 | 0 |
| 0 | 4 | 17 | 12 | 24 | 32 | 21 | 21 | 24 | 22 | 2 | 0 | 0 | 0 | 0 | 0 | 0 | 0 |
| 0 | 0 | 14 | 28 | 22 | 16 | 23 | 23 | 24 | 7 | 0 | 0 | 0 | 0 | 0 | 0 | 0 | 0 |
| 0 | 0 | 0 | 9 | 18 | 24 | 17 | 16 | 5 | 0 | 0 | 0 | 0 | 0 | 0 | 0 | 0 | 0 |
| 0 | 0 | 0 | 0 | 0 | 4 | 2 | 0 | 0 | 0 | 0 | 0 | 0 | 0 | 0 | 0 | 0 | 0 |
| 0 | 0 | 0 | 0 | 0 | 0 | 0 | 0 | 0 | 0 | 0 | 0 | 0 | 0 | 0 | 0 | 0 | 0 |
| 0 | 0 | 0 | 0 | 0 | 0 | 0 | 0 | 0 | 0 | 0 | 0 | 0 | 0 | 0 | 0 | 0 | 0 |
| 0 | 0 | 0 | 0 | 0 | 0 | 0 | 0 | 0 | 0 | 0 | 0 | 0 | 0 | 0 | 0 | 0 | 0 |
| 0 | 0 | 0 | 0 | 0 | 0 | 0 | 0 | 0 | 0 | 0 | 0 | 0 | 0 | 0 | 0 | 0 | 0 |

**Figure 2.** Data distribution of partition function *p* = 18.

| 0 | 41 | 68 | 15 | 0 | 0 | 0 |
|---|---|---|---|---|---|---|
| 36 | 158 | 154 | 111 | 0 | 0 | 0 |
| 79 | 141 | 160 | 133 | 15 | 0 | 0 |
| 65 | 155 | 143 | 143 | 5 | 0 | 0 |
| 8 | 128 | 148 | 82 | 0 | 0 | 0 |
| 0 | 2 | 10 | 0 | 0 | 0 | 0 |
| 0 | 0 | 0 | 0 | 0 | 0 | 0 |

**Figure 3.** Data distribution of partition function *p* = 7.

The dataset is divided by a partition function. If the data points in the datasets are evenly distributed in the partitioning area, the definition of threshold function T1 can be obtained, as shown in Formula (5).

$$T1 = \frac{|N|}{P * P} \tag{5}$$

Since the number of outliers is unknown in advance, it is impossible to judge correctly only by threshold function T1. Therefore, we propose the definition of threshold function T2.

$$T2 = |P| \tag{6}$$

The threshold partition function plays an important role in dataset optimization. If the threshold function value is too large, the dense data in the dataset will be over-preserved. If the threshold function value is small, the sparse datasets and even the outliers in the datasets are filtered out. Neither of them can achieve the effect of initialization filtering. We propose a threshold function that can be applied to many datasets.

Figure 4 shows the data volume comparison curve between the original dataset and the candidate dataset. Figure 4a shows the original datasets with data amounts of 998 and 3100. With the increase of the threshold function, the amount of data in candidate datasets increases gradually. Figure 4b shows the original datasets with data amounts of 8780 and 25,390. The number of outliers in the dataset is

unknown. If the number of outliers is large, the threshold function T1 of dataset average partition is too small. If the threshold function is too small, too many data points or even outliers will be initialized, which will affect the accuracy of the algorithm. When the threshold function is T2, the processing effect of the algorithm is better, and the accuracy of the algorithm is guaranteed while the initialization filtering is completed.

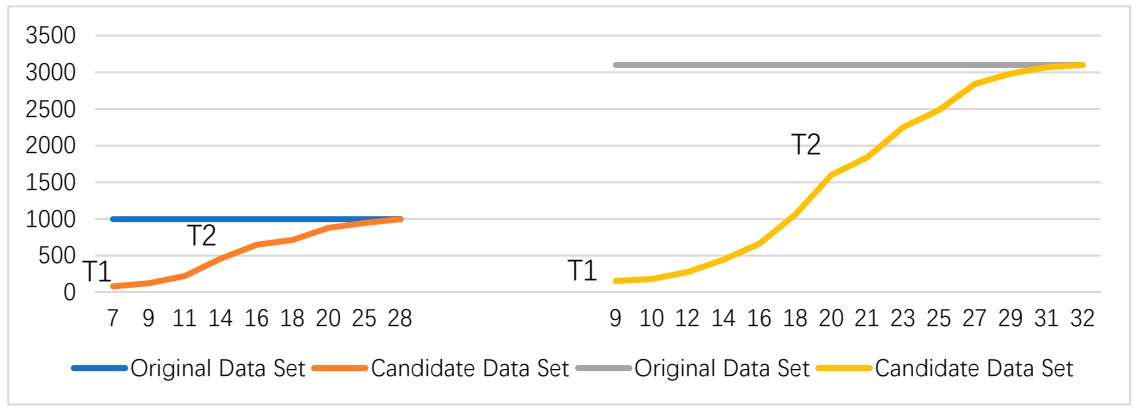

(**a**)

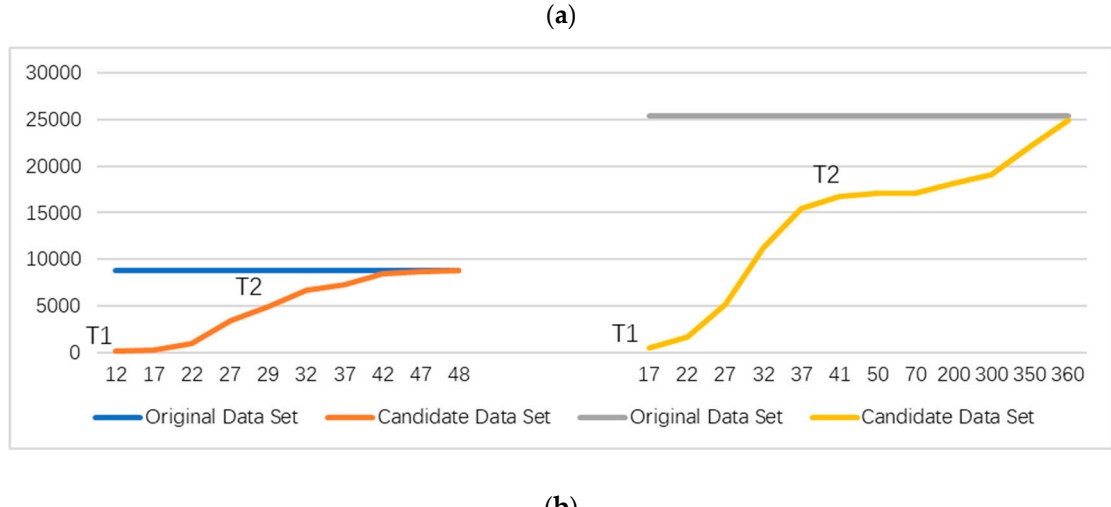

(**b**)

**Figure 4.** The effects of the size of threshold partitioning functions (T1 denotes the threshold function value is too small, and T2 is the threshold function value proposed in this paper) on candidate datasets: (**a**) describes the original datasets of |N| = 998 and |N| = 3100; (**b**) describes the original datasets of |N| = 8780 and |N| = 25,390.

According to our analysis, the definitions of the final partition function and the threshold function are obtained. We express it mathematically as follows.

**Definition 6.** *Partition function:*

$$P = |N|^{1/3} + |N|^{1/4}$$

**Definition 7.** *Threshold function:*

$$T2 = |P|$$

Naturally, as for the data set optimization method, there are cases where the apparent outliers in one partition may be close to the points in another partition. We divide it into three cases:

1.　General situation: Outliers are distributed far away from dense clusters as shown in Figure 1. The partition function and threshold function increase (decrease) with the increase (decrease) of the dataset. The purpose of partitioning is to exclude data points with very high density in the dataset, so outliers in the dataset will be retained by the data set optimization method.

2.　Special cases: Special outliers (such as individual outliers and a dense data cluster are closer), after partitioning non-dense data clusters and outliers are retained. Outliers are detected by a more accurate outlier detection algorithm in the next step.

3.　Very special case: Very special outliers (such as individual outliers and a dense data cluster are located very close). Most of the outlier detection algorithms cannot guarantee that all outliers can achieve complete detection, so the method proposed in our paper can take both detection efficiency and detection accuracy into account when the number of outliers is large and small. The next step will be a more in-depth study for the very special case.

### 3.2. Parameter-Free Outlier Detection

Outlier detection is an important aspect of data mining. The commonly used outlier detection methods are distance-based and density-based outlier detection methods, and their improved methods. These methods are based on the concept of nearest neighbor, so the nearest neighbor selection is very important for outlier detection. As shown in Figure 5, when the nearest neighbor k of data point o1 is small, the cluster that should be a cluster is divided into several data clusters, which results in the neglect of some neighborhood relationships that should exist. The representation of this situation in the nearest neighborhood graph is that the original whole connected subgraph is split into many small connected subgraphs, and then the data points with the same properties are misclassified into different clusters to get the wrong data relations. Similarly, if we set the nearest neighbor value larger, it will have an overall impact, so that the dataset itself is not a cluster that is divided into a data cluster. Secondly, on the whole, o2 is an outlier, but when the nearest neighbor of o1 is r, the outlier o2 is the nearest neighbor of the data point o1.

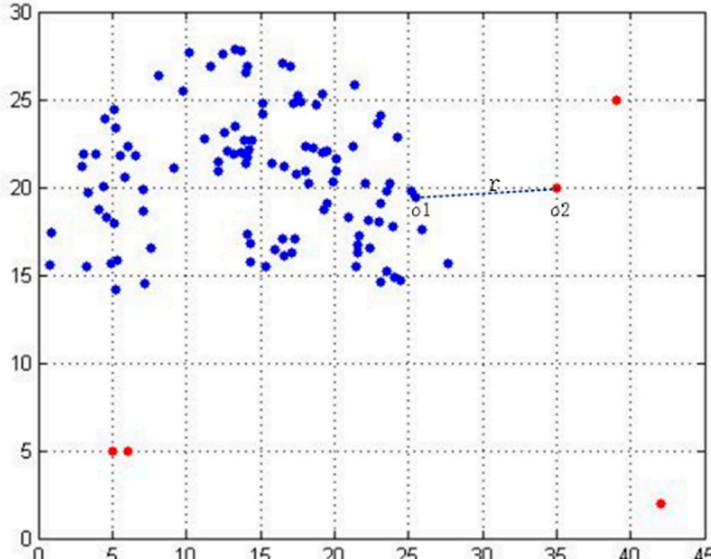

**Figure 5.** The effects of parameter selection in outlier detection algorithms (normal points in blue (o1), outliers point in red (o2)).

The nearest neighbor can reflect the relationship between data points, and one form of this relationship is the neighborhood graph, which can be constructed by connecting each data point to its nearest neighbor [25]. However, it is difficult to identify outliers accurately only through the single

concept of nearest neighbor, so the concept of mutual neighbor is introduced. The mutual neighbor graph can be constructed by connecting the mutual neighbors of each data point.

**Definition 8.** *Nearest neighbor search method. For a given dataset X and the data point q to be queried. The goal of the nearest neighbor search is to find a subset of Formula (1) in dataset X.*

$$NN(q, X) = \{x \in X \mid \forall p \in X : d(q, x) \leq d(p, q)\}. \tag{7}$$

**Definition 9.** *Mutual neighbor (MN). If p is the neighbor of q and q is the neighbor of p, then we call p and q mutual neighbors.*

**Definition 10.** *Mutual neighbor graphs. The graph constructed by connecting each point to its mutual neighbor is called a mutual neighbor graph.*

As shown in Figure 6, the mutual neighborhood graph consists of data points. Firstly, the first nearest neighbor of each data point is searched, and the arrow points to the nearest neighbor direction of the data point, and the nearest neighbor of each data point is obtained, at the same time, the mutual neighbor points are obtained. As shown in Figure 6a, O2 is the nearest neighbor of O3 and O3 is the nearest neighbor of O2, so O2 and O3 are mutual neighbors. Similarly, continuing to search for the second nearest neighbor of each data point, and getting the nearest neighbor and mutual neighbor of each data point. As shown in Figure 6b, the point without mutual neighbors is only O1. Then, we look for the third nearest neighbor of the data point, as shown in Figure 6c. At this point, there is still only O1 without mutual neighbors. When the nearest neighbors k = 2 and k = 3, the number of points without mutual neighbors is the same, and they are only O1 points, so the mutual neighbor graph of the whole dataset remains stable and the algorithm ends. As the number of nearest neighbors increases gradually, the number of mutual neighbors of normal data points increases correspondingly, so that the mutual neighbor graph becomes more and more complex, and the outliers remain unchanged. Based on the above description, we can judge the abnormal situation of data points in the dataset by observing the changes of the mutual neighbor graph of data points.

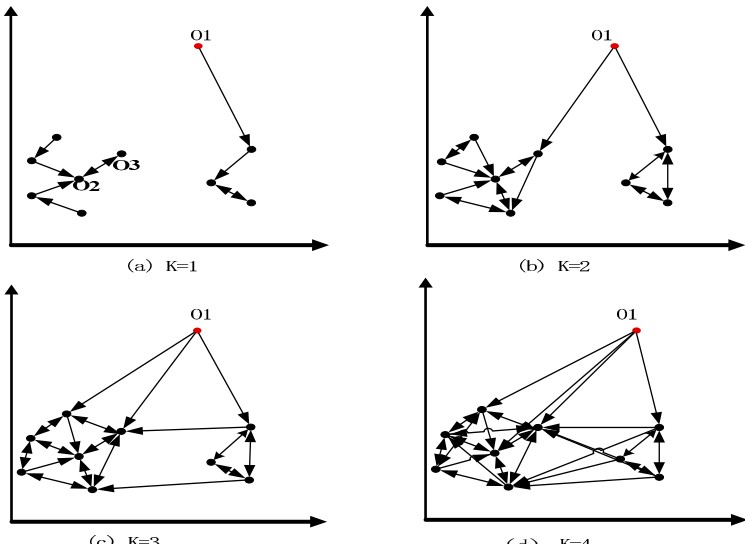

**Figure 6.** With the gradual increase of k, the changes of the mutual neighbor graph of the data points.

As the mutual neighbor graph is a form of a data neighbor graph, for many datasets with complex distribution, it cannot be measured accurately through a mutual neighbor graph intuitively, so we

propose the concept of the number of residual neighbors in this paper. With the gradual increase of the nearest neighbor k, the number of mutual neighbors of data points is constantly changing; meanwhile, the number of data points without mutual neighbors is also changing. To make outlier detection more intuitive and effective, we can judge the outliers by observing the change of the number of residual neighbors.

**Definition 11.** *The number of residual neighbors. If the data points p and q are mutual neighbors, the number of data points without such a mutual neighborhood is the number of residual neighbors.*

As shown in Figure 7, it is a variation curve of the number of residual neighbors of the data point in Figure 6. It can be seen intuitively that when k = 2 and k = 3, the number of residual neighbors is equal, and the number of residual neighbors in the dataset is 1, which satisfies the termination condition of the algorithm. Therefore, it is considered that the whole neighborhood is stable and the algorithm is terminated. So, for the dataset shown in Figure 6, O1 is the outlier obtained by the parameter-free outlier detection algorithm.

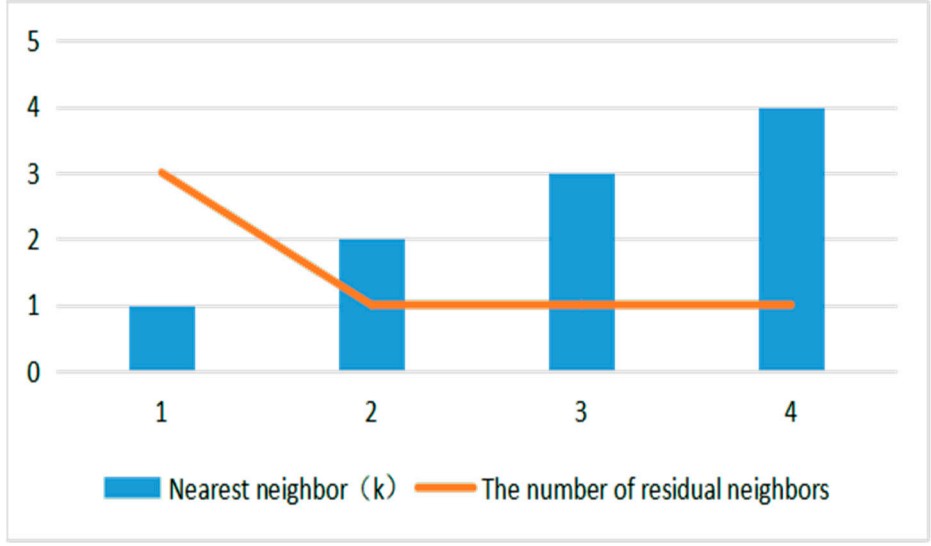

**Figure 7.** With the gradual increase of k, the changes of the residual neighbors of the data points.

In some distributions of data, outliers exist not only in the form of single points but also in the form of outliers. As shown in Figure 8, O1 and O4 are outliers and the rest are normal points. Firstly, as the number of nearest neighbors increases gradually, the mutual neighbor graphs of dense clusters become more and more complex until they are stable (only k = 1–4 mutual neighbor graphs are listed in this paper). Secondly, we get the number of residual neighbors as shown in Figure 9. When k = 2 and k = 3, the number of residual neighbors is equal, and the number of residual neighbors in the dataset is 0, which satisfies the termination condition of the algorithm, but the outliers O1 and O4 of the algorithm cannot be detected correctly.

Therefore, in order to improve the accuracy of the algorithm, when the relationship between data clusters is stable, we compare the size of each data cluster and the number of nearest neighbors. If the size of the cluster is smaller than the number of nearest neighbors, it means that the data cluster is an anomalous cluster, so we can obtain a more accurate set of outliers.

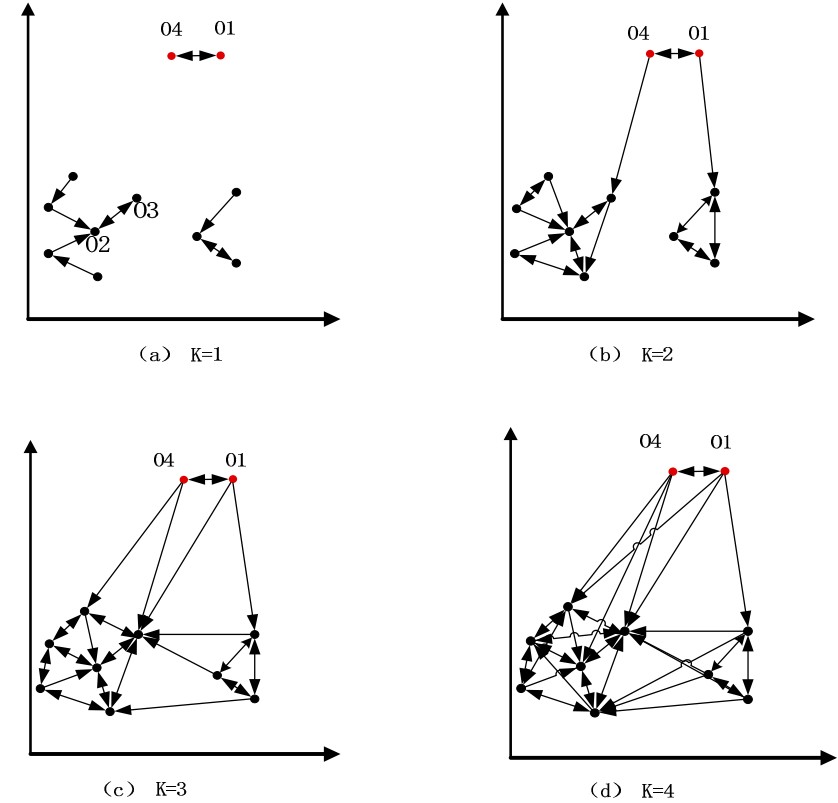

**Figure 8.** With the gradual increase of k, the changes of the mutual neighbor graph of the data points.

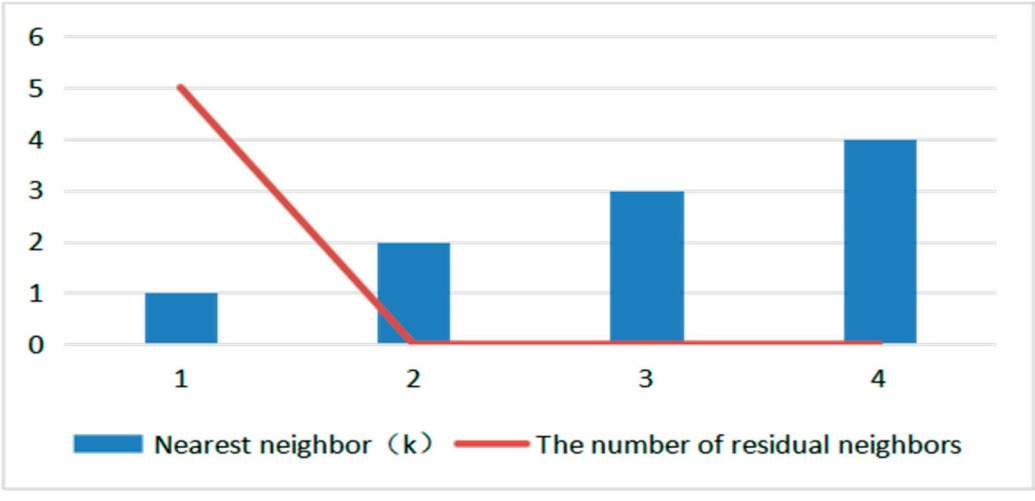

**Figure 9.** With the gradual increase of k, the changes of the residual neighbors of the data points.

Based on the concept of the number of residual neighbors, Algorithm 2 is used to describe the process of parameter-free outlier detection. In Algorithm 2, we first input the initialized dataset. As the number of nearest neighbors k increases gradually, we traverse to calculate the number of mutual neighbors and residual neighbors until the number of residual neighbors does not change. Then, a single outlier is obtained. By comparing the size of the data cluster with the nearest neighbor k, we get the outlier cluster and the outlier set as follows:

---

**Algorithm 2.** Parameter-free outlier detection

---

Input: Candidate dataset
Output: Outlier Dataset
Constructing the Mutual Neighbor Graph G; k; C1;
k = 1;
(1)Random an object x; visited(x) = true;
   while y∈G ∪ visited(y)! = true
then visited(y) = true; calculate (y, mutual neighbor);
Calculate (y, single neighbor, num)
k = 2; go to(1);
k = 3; go to(1);
… …
if num then outlier = true;
if(C1<k)
   then outlier = true

---

There are several characteristics of data distribution in outlier detection: the number of normal data points is large and dense, the number of abnormal data points is small and sparse, and the number of nearest neighbors is uncertain. Firstly, we study the shortcomings of the existing outlier detection algorithms, then combine the above data distribution characteristics, and finally propose the parameter-free outlier detection algorithm in our paper. The algorithm realizes the function of detecting more reasonable outliers efficiently without human participation.

## 4. Experiments

We compared the performance of our proposed parameter-free outlier detection algorithm in three aspects. The first part verifies the effectiveness of DOM dataset optimization method in outlier detection by comparing the sensitivity of dataset partitioning. In the second part, it is proved that the proposed parameter-free outlier detection algorithm outperforms the density-based outlier detection algorithm in terms of efficiency by comparing the running time of them. The third part demonstrated the superiority of our model by comparing the AUC of the parameter-free outlier detection algorithm and the density-based outlier detection algorithm. To conform to the distribution rule of the dataset, we deleted some objects in the dataset and added them as outliers. The algorithm is tested on a 12-core Intel Xeon 3.5 GHz CPU, 32-G RAM 64-bit Windows 10 and presented in VS2010 and MATLAB R2014a.

### 4.1. Description of Datasets

In our paper, we chose the commonly used dataset in outlier detection. In order to verify the accuracy and rationality of the experiment, we verified it via using the datasets of different scales and types. Table 1 shows the datasets of |N| = 213–100,000 and Table 2 shows the five real-world datasets.

Table 1 shows the overview of 10 datasets, in which about 1–4% of outliers are generated randomly. Table 2 presents the overview of five common datasets, in which 1–9% of the abnormal values are generated randomly.

**Table 1.** The summary of 10 datasets (|**N**| = 213–100,000).

| No. of samples | 213 | 998 | 3100 | 5632 | 8783 |
|---|---|---|---|---|---|
| No. of outliers | 4 | 34 | 60 | 101 | 198 |
| No. of samples | 16,984 | 25,395 | 58,654 | 81,324 | 100,000 |
| No. of outliers | 152 | 995 | 1267 | 1426 | 1000 |

**Table 2.** The summary of 5 real-world datasets.

| Dataset | Liver Disorder | Wholesale | Aggregation | Wine-Quality | Page-Blocks |
|---|---|---|---|---|---|
| No. of samples | 240 | 400 | 788 | 4828 | 5453 |
| No. of outliers | 8 | 78 | 7 | 131 | 487 |

### 4.2. Effectiveness of Dataset Optimization Method

The criterion for evaluating the effectiveness of dataset optimization methods is to filter out normal data points and ensure that outliers are among candidate datasets. That is sensitivity. If the outliers are filtered out due to dataset optimization, the dataset optimization method is invalid. The calculation formula of sensitivity is as follows.

$$\text{sensitivivity} = \frac{\text{outliers in candidate samples}}{\text{outliers}} \times 100\%. \tag{8}$$

If the sensitivity of DOM filtered is 100%, the candidate datasets are sensitive to all outliers, which proves that DOM is effective at this time.

We verified the validity of DOM method through multiple datasets. Tables 3 and 4 show the datasets we selected, the number of candidate samples after initialization, and the number of outliers in the samples. By comparing the number of samples and outliers in the original dataset, the number of samples and outliers in the candidate samples. We can observe that, first of all, DOM reduces the size of the data and ensures that outliers still exist in candidate datasets. Secondly, the dataset optimization method obtains candidate datasets. At this time, the normal data points in candidate datasets are still obvious compared with outliers, which does not affect the next step of parameter-free outlier detection. The sensitivity of candidate datasets to outliers is 100%, so the validity of DOM is proved effectively. The experimental results (sensitivity) show that the DOM dataset optimization method proposed in this paper is suitable for both large and small datasets. The reason is that it can reduce the amount of data on the basis of ensuring the accuracy of the algorithm and provide a more efficient operation prerequisite for the next outlier detection algorithm.

**Table 3.** The summary of 10 datasets ($|\mathbf{N}|$ = 213–100,000) after dataset optimization.

| No. of Samples | No. of Outliers | Partition Function | No. of Candidate Samples | No. of Outliers in Candidate Samples | Sensitivity |
|---|---|---|---|---|---|
| 213 | 4 | 8 | 145 | 4 | 100% |
| 998 | 34 | 14 | 601 | 34 | 100% |
| 3100 | 60 | 21 | 1736 | 60 | 100% |
| 5632 | 101 | 25 | 2623 | 101 | 100% |
| 8783 | 198 | 29 | 3443 | 198 | 100% |
| 16,984 | 152 | 36 | 8904 | 152 | 100% |
| 25,395 | 995 | 41 | 13,712 | 995 | 100% |
| 58,654 | 1267 | 53 | 16,708 | 1267 | 100% |
| 81,324 | 1426 | 59 | 23,459 | 1426 | 100% |
| 100,000 | 1000 | 63 | 31,523 | 1000 | 100% |

**Table 4.** The summary of 5 real-world datasets after dataset optimization.

| Dataset | No. of Samples | No. of Outliers | No. of Candidate Samples | No. of Outliers in Candidate Sample | Sensitivity |
|---|---|---|---|---|---|
| Liver disorder | 240 | 8 | 154 | 8 | 100% |
| Wholesale | 400 | 78 | 261 | 78 | 100% |
| Aggregation | 788 | 7 | 398 | 7 | 100% |
| Wine-Quality | 4828 | 131 | 2462 | 131 | 100% |
| Page-Blocks | 5453 | 487 | 2674 | 487 | 100% |

### 4.3. Operating Efficiency of Algorithms

We tested our algorithm in several datasets of different scales and dimensions and compared the detection efficiency of the algorithm. Running time is the criterion for evaluating the efficiency of the algorithm. For the same dataset, the less time the algorithm consumes, the better the efficiency of the algorithm, and the more time the algorithm consumes, the worse the efficiency of the algorithm. Figure 10a shows a number of datasets with a data size of 213–100,000. Figure 10b is five common datasets. The abscissa refers to the type and size of each dataset, and the ordinate denotes the time consumed by the algorithm(s). The consumption time of dataset (|N| = 213–8783) is shown in the right ordinate of Figure 10a. The consumption time of dataset (|N| = 16,984–100,000) is shown on the left side of Figure 10a. As the data size increases, the time consumed by the algorithm increases. Conversely, the proposed parameter-free outlier detection algorithm consumes less time than the density-based outlier detection algorithm. In this section, we use the running time(s) for comparison. Figure 10a,b show comparison of the running time between the parameter-free outlier detection algorithm and the density-based outlier detection algorithm based on the same dataset. It can be concluded that the proposed parameter-free outlier detection algorithm consumes less time and has higher efficiency of outlier detection. Besides, with the increase in dataset size, the superiorities of our proposed parameter-free outlier detection algorithm become more obvious. The experimental results show that, compared with the density-based algorithm, our DOM-based parameter-free outlier detection algorithm reduces the running time of the algorithm and improves the efficiency of the algorithm, especially in large datasets.

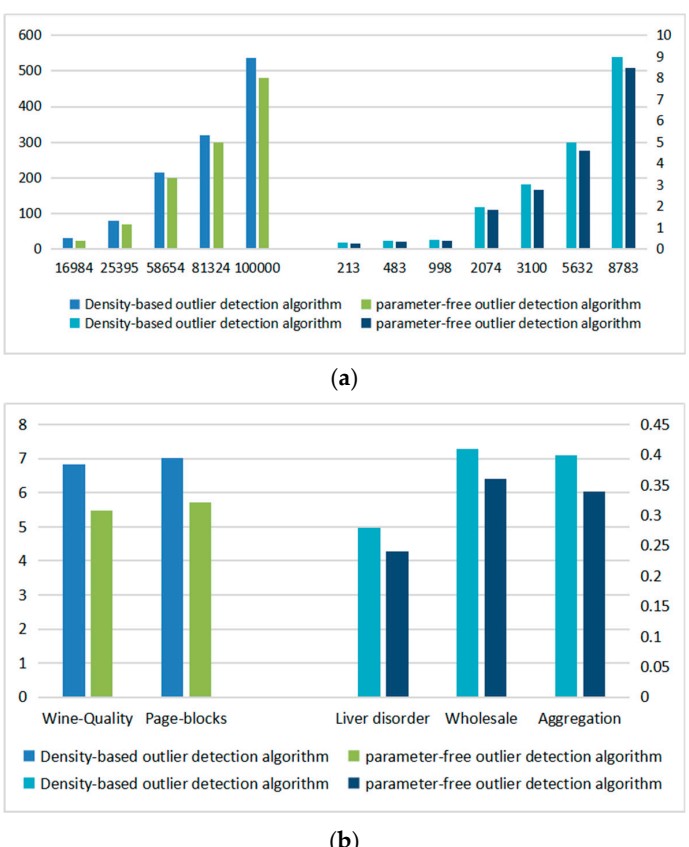

**Figure 10.** Comparisons of algorithm time consumption: (**a**) Comparisons of the time consumed by a parameter-free outlier detection algorithm and density-based outlier detection algorithm(s) (the size of datasets: |N| = 213–|N| = 100,000); (**b**) comparisons of the time consumed by a parameter-free outlier detection algorithm and density-based outlier detection algorithm(s) (real-world datasets).

### 4.4. Efficiency of Algorithms

In this section, we tested our algorithm in 10 datasets of different sizes and 5 standard datasets. The performance of parameter-free outlier detection algorithm and density-based outlier detection algorithm(s) is evaluated.

Figure 11a shows the distribution of datasets with data size |N| = 2073. It contains a data cluster composed of several normal data points, as well as a number of outliers distributed on the edge. The different colors in Figure 11 are the result of overlapping data points.

With the gradual increase of the value of partition function and threshold function, the data partition judgment results, as shown in Figure 11b–e, are obtained.

Figure 11b,c shows the result distribution when the partition function and threshold function are small. Figure 11d is the result distribution of our algorithm. Figure 11e shows the result distribution when the partition function and threshold function are large.

As shown in Figure 11b, when the value of partition function and threshold function are very small, the dataset optimization method is not only difficult to play a role, but also filters out some abnormal values, which affects the detection accuracy of the final algorithm. As shown in Figure 11e, when the value of partition function and threshold function are large, the dataset optimization method cannot initialize the dataset and improve the efficiency of the algorithm. Figure 11d is the result distribution of our algorithm. The experimental results show that the sensitivity of outlier detection algorithm is not affected, the DOM method proposed in this paper can filter the high-density data clusters in the dataset and further reduce the amount of data to a certain extent. It also provides a more effective premise for the next step of an outlier detection algorithm.

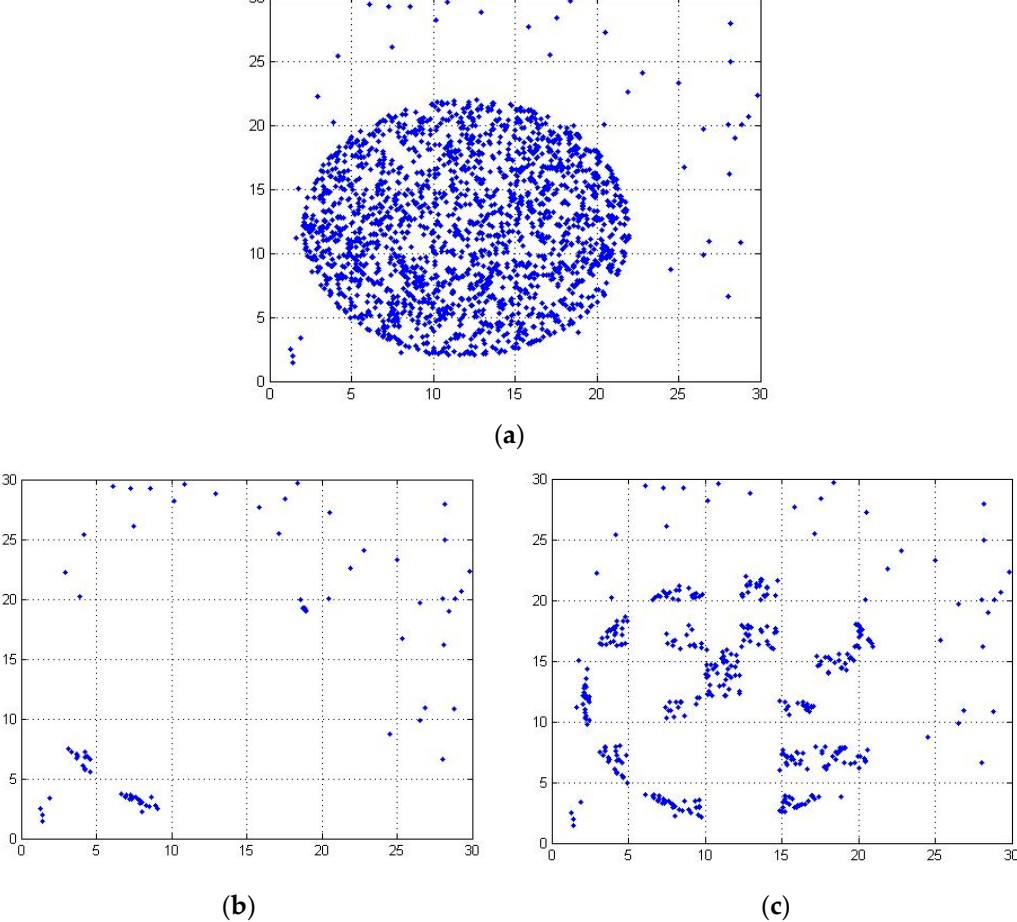

(a)

(b)                                             (c)

**Figure 11.** *Cont.*

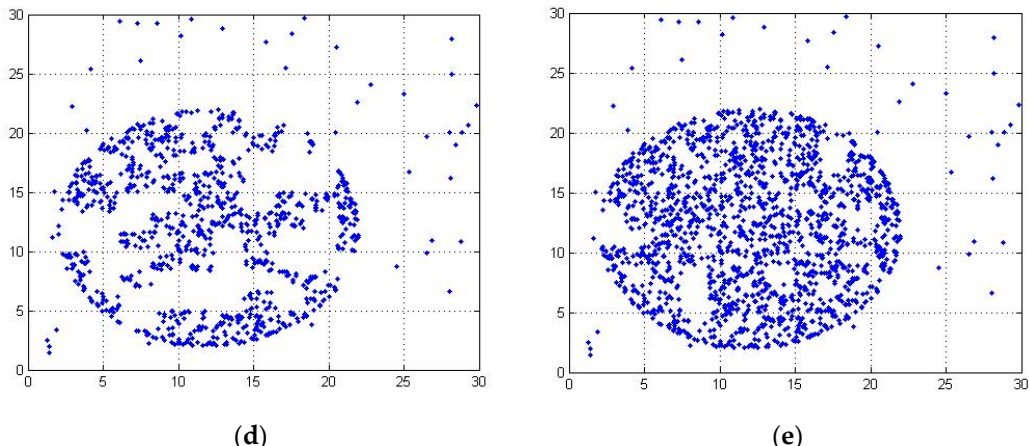

(**d**)                                              (**e**)

**Figure 11.** Experimental comparison of the effectiveness of our DOM method: (**a**) Dataset distribution with |N| = 2073; (**b**) dataset distribution after being divided by P,T (|N| = 81); (**c**) dataset distribution after being divided by P,T (|N| = 431); (**d**) dataset distribution after being divided by P,T (|N| = 1014); (**e**) dataset distribution after being divided by P,T (|N| = 1729).

Figure 12a shows the distribution of datasets with a data size of |N| = 100,000. It contains a data cluster composed of multiple normal data points, as well as multiple outliers. The different colors in Figure 12 are the result of overlapping data points.

Figure 12b–g are the result distribution map obtained by partition function and threshold function to partition and judge the dataset. The samples included are 4822, 6162, 14,010, 24,512, 31,523 and 79,051 respectively. The traditional outlier detection method needs to calculate the outliers of all data points. The DOM method proposed in this paper can filter the high-density data clusters in datasets. As shown in Figure 12f, it is the results distribution diagram of our proposed parameter-free outlier detection algorithm. Through the DOM dataset optimization method proposed by our paper, the original dataset with |N| = 100,000 is optimized to a dataset with |N| = 31,523, on the premise that the sensitivity of the dataset is not affected. DOM filters four blue clusters of high-density data.

Experimental results show that our DOM dataset optimization method shows particularly good performance especially in large datasets, which can reduce the amount of data to a certain extent, and provide a more effective premise for the next step of an outlier detection algorithm.

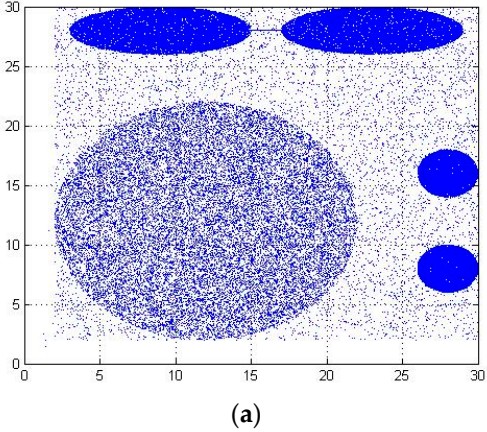

(**a**)

**Figure 12.** *Cont.*

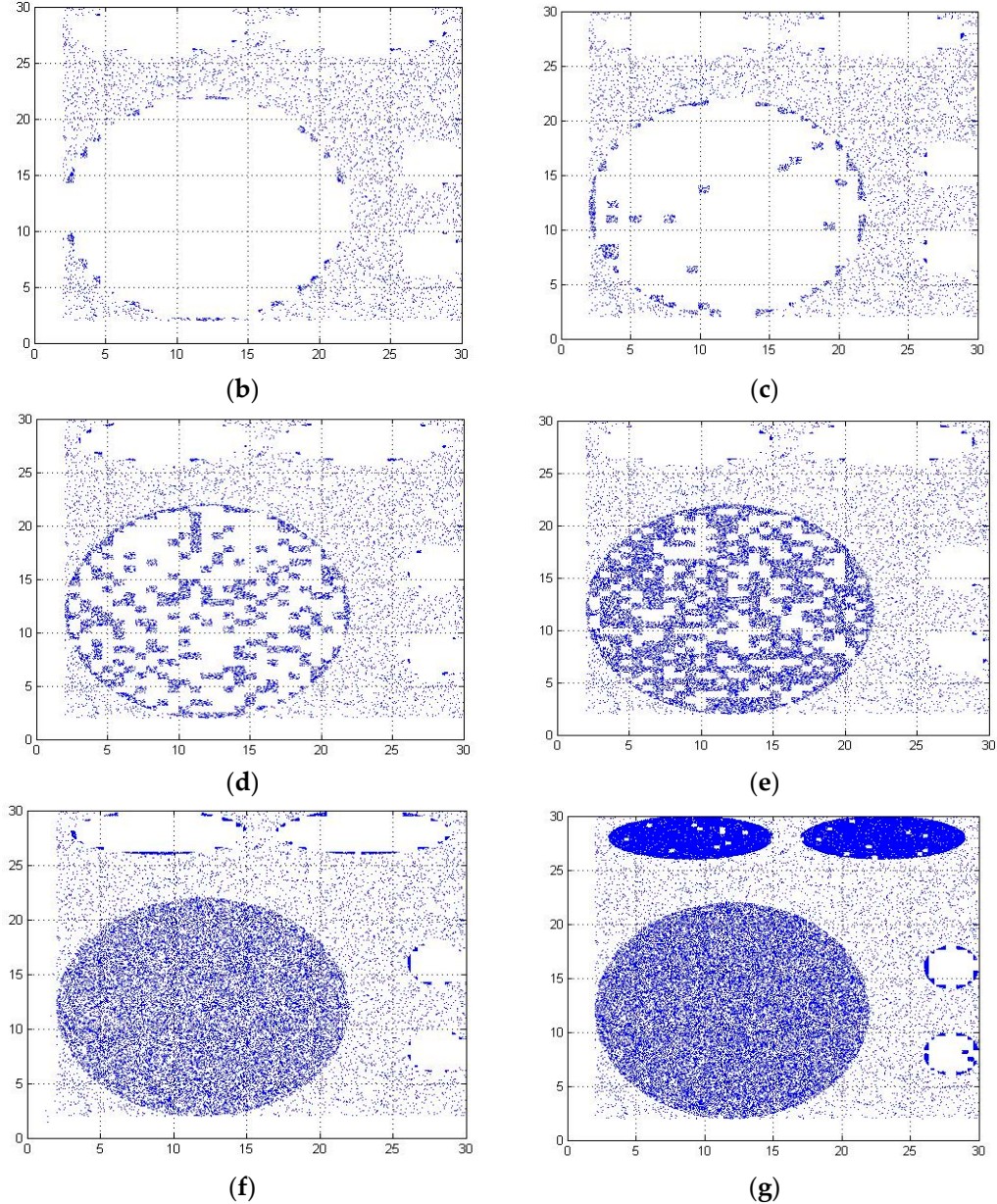

**Figure 12.** Experimental comparison of the effectiveness of our DOM method: (**a**) Distribution of the original dataset (|N| = 100,000); (**b**) dataset distribution after being partitioned by P,T (|N| = 4822); (**c**) dataset distribution after being partitioned by P,T (|N| = 6162); (**d**) dataset distribution after being partitioned by P,T (|N| = 14,010); (**e**) dataset distribution after being partitioned by P,T (|N| = 24,512); (**f**) dataset distribution after being partitioned by P,T (|N| = 31,523); (**g**) dataset distribution after being partitioned by P,T (|N| = 79,051).

Taking a small real-world dataset with data size 788 as an example, the effectiveness of the DOM method is verified. Figure 13a shows the distribution of aggrestration datasets. In Figure 13a, the blue data points represent 781 normal data points and the red data points represent 7 outliers added manually. Figure 13b shows the results of partitioning and judging aggrestration datasets using partitioning functions and threshold functions. By using our proposed DOM method, high-density data clusters in datasets can be filtered. At this point, the seven outliers manually added still exist in the candidate dataset (the sensitivity of the dataset is 100%). The experimental results show that our DOM dataset optimization method also improves the performance of outlier detection in small datasets.

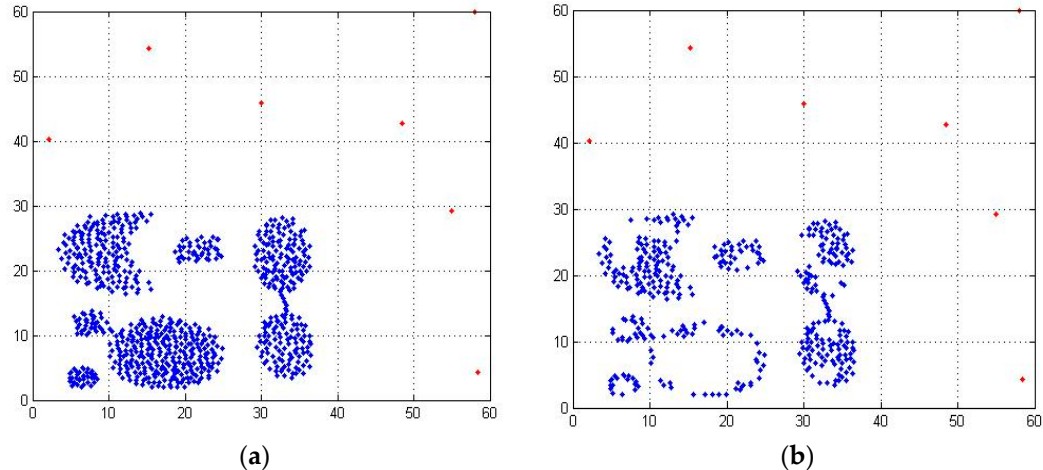

**Figure 13.** Contrast chart of aggrestration dataset processing with DOM: (**a**) Distribution of original aggrestration dataset; (**b**) distribution of candidate aggrestration dataset.

In order to evaluate the validity of the model, AUC (Area Under the Curve) is a reliable classification evaluation index, which indicates the area under the ROC (Receiver Operating Characteristic) curve. The horizontal coordinate of the ROC curve is false positive FPR, and the vertical coordinate is true positive TPR. The value range of AUC is 0 to 1. The larger the AUC, the better the effect of an outlier detection algorithm. The relevant formulas are as follows:

$$FPR = \frac{FP}{FP + TN}. \tag{9}$$

$$TPR = \frac{TP}{TP + FN}. \tag{10}$$

Definitions are as follows:

True Positive (TP): Predict normal samples as normal classes;
True Negative (TN): Predict the number of abnormal samples as the number of abnormal classes;
False Positive (FP): Predict abnormal samples as normal classes;
False Negative (FN): Predict normal samples as abnormal classes.

Table 5 shows the results of the outlier detection algorithm for outlier partitioning of different scale manual datasets. P-F O D Algorithm refers to the parameter-free outlier detection algorithm; D-based O D Algorithm refers to the density-based outlier detection algorithm. The density-based outlier detection algorithm selects the nearest neighbor range between k = 1–16. Compared with density-based outlier detection algorithm, the AUC value of parameter-free outlier detection algorithm is higher, and the effect of outlier detection algorithm is better. Through experiments, we can observe that the parameter-free outlier detection algorithm can filter high-density datasets compared with the density-based outlier detection algorithm, ensuring that the accuracy of the algorithm is not affected.

Table 6 shows the results of outlier detection for five real-world datasets. The nearest neighbor range of the density-based outlier detection algorithm is k = 1–9. Compared with density-based outlier detection algorithm, the AUC value of parameter-free outlier detection algorithm is higher, and the effect of outlier detection algorithm is better. For the aggregation dataset in Table 6, because the number of outliers inserted manually is small and the outlier features are obvious, both the parameter-free outlier detection algorithm and the density-based algorithm can effectively detect the outliers. However, the parameter-free outlier detection algorithm greatly reduces the running time of the algorithm. Therefore, our method reduces the time complexity and ensures the accuracy of the algorithm, which further improves the comprehensive performance of the algorithm.

Therefore, we can conclude that the proposed parameter-free outlier detection algorithm improves the efficiency of the algorithm on the premise of ensuring the accuracy of the algorithm, especially in large datasets.

**Table 5.** Detection of datasets by different outlier detection algorithms.

| No. of Samples | No. of Outliers | P-F O D Algorithm (AUC) | D-Based O D Algorithm (AUC) |
|---|---|---|---|
| 213 | 4 | 0.884 | 0.862 |
| 998 | 7 | 0.978 | 0.946 |
| 3100 | 21 | 0.984 | 0.951 |
| 5632 | 45 | 0.962 | 0.943 |
| 8783 | 67 | 0.968 | 0.944 |
| 16,984 | 90 | 0.988 | 0.956 |
| 25,395 | 102 | 0.994 | 0.965 |
| 58,654 | 152 | 1.000 | 1.000 |
| 81,324 | 221 | 1.000 | 0.989 |
| 100,000 | 516 | 0.972 | 0.949 |

**Table 6.** Detection of real-world datasets by different outlier detection algorithms.

| Datasets | No. of Samples | No. of Outliers | P-F O D Algorithm (AUC) | D-Based O D Algorithm (AUC) |
|---|---|---|---|---|
| Live disorder | 240 | 8 | 0.671 | 0.669 |
| Wholesale | 400 | 108 | 0.765 | 0.778 |
| Aggregation | 788 | 7 | 1.000 | 1.000 |
| Wine-Quality | 4828 | 131 | 0.769 | 0.771 |
| Page-Blocks | 5453 | 487 | 0.691 | 0.702 |

## 5. Conclusions and Future Work

We deeply studied the shortcomings of the existing outlier detection algorithms and proposed a parameter-free outlier detection algorithm in this paper. Through the dataset optimization method, we completed the initialization of the dataset and reduced the time complexity of the algorithm. Based on the concept of the number of remaining neighbors proposed by us, this paper implements the nonparametric outlier test. In practical application, it has a good effect for outlier detection, especially for outlier detection of big datasets. In future work, we will continue to explore other factors that affect the accuracy of this algorithm and the practical application of this algorithm.

**Author Contributions:** Conceptualization, L.W. and L.S.; funding acquisition, L.S., P.L. and L.X.; methodology, L.W.; project administration, L.X.; resources, P.L. and L.X.; software, L.W.; validation, L.W.; visualization, L.W.; writing—original draft, L.W.; writing—review and editing, L.X., Y.D. and L.Z. All authors have read and agreed to the published version of the manuscript.

**Funding:** This work was supported by the National Social Science Fund (19BYY076), the Science Foundation of the Ministry of Education of China (no. 14YJC860042), and the Shandong Provincial Social Science Planning Project (no. 19BJCJ51/18CXWJ01/18BJYJ04/16CFXJ18/16CXWJ01).

**Acknowledgments:** Thanks to all commenters for their valuable and constructive comments.

**Conflicts of Interest:** The authors declare no conflict of interest.

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
