# Peer review of "A Parameter-Free Outlier Detection Algorithm Based on Dataset Optimization Method"

_information, doi:10.3390/info11010026_

Round 1
Reviewer 1 Report
Well this is a good manuscript to read, also it is interesting.
However, I have few minor comments on this.
1) Two first items in abstract
2) Need a detailed comparison of traditional and innovative outlier detection algorithms
3) Can you introduce two color codes for C1 and C2?
4) I think, the reader is lost for finding algorithm 1 while reading figure 1. Therefore, re-structuring is needed.
5) This is not a conclusion. It is a summary. What did u find fro this reasearch. That is what we write in conclusions.
Reviewer 2 Report
The problem addressed is important, however the technical novelty is limited.
The initialization part where the high-density points are dropped, is trivial. There is no proof of correctness of the partition and threshold steps. After partitioning the points, will the detected outliers be the same as without partitioning? How is that ensured, as an apparent outlier in one partition may be close to the points in another partition (so not an outlier)? The references are mostly text books, and the recent research literature is missing. For example, the following two references are highly related –a. Rémi Domingues, Maurizio Filippone, Pietro Michiardi, Jihane Zouaoui, A comparative evaluation of outlier detection algorithms: Experiments and analyses, Pattern Recognition, Volume 74, 2018, Pages 406-421. b. Blouvshtein and D. Cohen-Or, "Outlier Detection for Robust Multi-Dimensional Scaling," in IEEE Transactions on Pattern Analysis and Machine Intelligence, vol. 41, no. 9, pp. 2273-2279, 2019. The set of 10 datasets for the experiment is extensive.Author Response
Please see the attachment.

Reviewer 3 Report
This paper proposes a new approach to outlier detection. The paper has technical merit, however, the English should be revised thoroughly before publication. For instance, in the Abstract: "...is to identify outliers in datasets and extract potential information in the datasets." --> "...is to identify outliers and to extract potential information in the datasets." and many more.
Round 2
Reviewer 2 Report
The authors have addressed most of my comments in the revised manuscript, including the concerns on technical depth, references, and methodology. Regarding my question "How is that ensured, as an apparent outlier in one partition may be close to the points in another partition (so not an outlier)?", the authors have explained different scenarios in their response letter, but those scenarios are not included in the revised manuscript. I suggest that they include their response of the three scenarios in the paper (preferably after the partition function is introduced).
